# Long-term Forecasting with TiDE: Time-series Dense Encoder

**Abhimanyu Das**                                             *abhidas@google.com*

*Google Research*

**Weihao Kong**                                             *weihaokong@google.com*
*Google Research*

**Andrew Leach**                                             *andrewleach@google.com*
*Google Cloud*

**Shaan Mathur**                                             *shaanmathur@google.com*
*Google Cloud*

**Rajat Sen**                                             *senrajat@google.com*
*Google Research*

**Rose Yu**                                             *q6yu@ucsd.edu*
*University of California, San Diego*

**Reviewed on OpenReview:** *https://openreview.net/forum?id=pCbC3aQB5W*

## Abstract

Recent work has shown that simple linear models can outperform several Transformer based approaches in long term time-series forecasting. Motivated by this, we propose a Multi-layer Perceptron (MLP) based encoder-decoder model, Time-series Dense Encoder (TiDE), for long-term time-series forecasting that enjoys the simplicity and speed of linear models while also being able to handle covariates and non-linear dependencies. Theoretically, we prove that the simplest linear analogue of our model can achieve near optimal error rate for linear dynamical systems (LDS) under some assumptions. Empirically, we show that our method can match or outperform prior approaches on popular long-term time-series forecasting benchmarks while being 5-10x faster than the best Transformer based model.

## 1 Introduction

Long-term forecasting, which is to predict several steps into the future given a long context or look-back, is one of the most fundamental problems in time series analysis, with broad applications in energy, finance, and transportation. Deep learning models (Wu et al., 2021; Nie et al., 2022) have emerged as a popular approach for forecasting rich, multivariate, time series data, often outperforming classical statistical approaches such as ARIMA or GARCH (Box et al., 2015). In several forecasting competitions such as the M5 competition (Makridakis et al., 2020) and IARAI Traffic4cast contest (Kreil et al., 2020), almost all the winning solutions are based on deep neural networks.

---

Author names are arranged in alphabetical order of last names.

Various neural network architectures have been explored for forecasting, ranging from recurrent neural networks to convolutional networks to graph neural networks. For sequence modeling tasks in domains such as language, speech and vision, Transformers (Vaswani et al., 2017) have emerged as the most successful model, even outperforming recurrent neural networks (LSTMs)(Hochreiter and Schmidhuber, 1997). Subsequently, there has been a surge of Transformer-based forecasting papers (Wu et al., 2021; Zhou et al., 2021; 2022) in the time-series community that have claimed state-of-the-art (SoTA) forecasting performance for long-horizon tasks. However, recent work (Zeng et al., 2023) has shown that these Transformerss-based architectures may not be as powerful as one might expect for time series forecasting, and can be easily outperformed by a simple linear model on forecasting benchmarks. Such a linear model however has deficiencies since it is ill-suited for modeling non-linear dependencies among the time-series sequence and the time-independent covariates. Indeed, a very recent paper (Nie et al., 2022) proposed a new Transformer-based architecture that obtains SoTA performance for deep neural networks on the standard multivariate forecasting benchmarks.

In this paper, we present a simple and effective deep learning architecture for long-term forecasting that obtains superior performance when compared to existing SoTA neural network based models on the time series forecasting benchmarks. Our Multi-Layer Perceptron (MLP)-based model is embarrassingly simple without any self-attention, recurrent or convolutional mechanism. Therefore, it enjoys a linear computational scaling in terms of the context and horizon lengths unlike many Transformer-based solutions.

The *main contributions* of this work are as follows:

- We propose the T̲ime-series D̲ense E̲ncoder (TiDE) model architecture for long-term time series forecasting. TiDE encodes the past of a time-series along with covariates using dense MLPs and then decodes time-series along with future covariates, again using dense MLPs.

- We analyze a simplified linear analogue of our model and prove that this linear model can achieve near optimal error rate in linear dynamical systems (LDS) (Kalman, 1963) when the design matrix of the LDS has maximum singular value bounded away from 1. We empirically verify this on a simulated dataset where the linear model outperforms LSTMs and Transformers.

- On popular real-world long-term forecasting benchmarks, our model achieves better or similar performance compared to prior neural network based baselines (>10% lower Mean Squared Error on the largest dataset). At the same time, TiDE is 5x faster in terms of inference and more than 10x faster in training when compared to the best Transformer based model.

## 2 Background and Related Work

Models for long-term forecasting can be broadly divided into either *multivariate* models or *univariate* models.

Multivariate models use the past of all the interrelated time-series variables and predict the future of all the time-series as a joint function of those pasts. This includes the classical VAR models (Zivot and Wang, 2006). We will mostly focus on the prior work on neural network based models for long-term forecasting. LongTrans (Li et al., 2019a) uses attention layer with LogSparse design to capture local information with near linear space and computational complexity. Informer (Zhou et al., 2021) uses the ProbSparse self-attention mechanism to achieve sub-quadratic dependency on the length of the context. Autoformer (Wu et al., 2021) uses trend and seasonal decomposition with sub-quadratic self attention mechanism. FEDFormer (Zhou et al., 2022) uses a frequency enchanced structure while Pyraformer (Liu et al., 2021) uses pyramidal self-attention that has linear complexity and can attend to different granularities. The common theme among the above works is the use of sub-quadratic approximations for the full self attention mechanism, that have been also been used in other domains (Wang et al., 2020).

On the other hand, univariate models predict the future of a time-series variable as a function of only the past of the same time-series and covariate features. In other words, the past of other time-series is not part of the input during inference. There are two kinds of univariate models, *local* and *global*. Local univariate

models are usually trained per time-series variable and inference is done per time-series as well. Different variables have different models. Classical models like AR, ARIMA, exponential smoothing models (McKenzie, 1984) and the Box-Jenkins methodology (Box and Jenkins, 1968) belong in this category. We would refer the reader to (Box et al., 2015) for an in depth discussion of these methods.

Global univariate models ignore the variable information and train one shared model for all the time series on the whole dataset. This category mainly includes deep learning based architectures like (Salinas et al., 2020). In the context of long-term forecasting, recently it was observed that a simple linear global univariate model can outperform the transformer based multivariate approaches for long-term forecasting (Zeng et al., 2023). DLinear (Zeng et al., 2023) learns a linear mapping from context to horizon, pointing to deficiencies in sub-quadratic approximations to the self-attention mechanism. Indeed, a very recent model, PatchTST (Nie et al., 2022) has shown that feeding contiguous patches of time-series as tokens to the vanilla self-attention mechanism can beat the performance of DLinear in long-term forecasting benchmarks. MLPs have been used for time-series forecasting in the popular N-BEATS model (Oreshkin et al.) and later extended in a follow up work (Challu et al., 2023) that uses multi-rate sampling for better efficiency. However, these methods do not explicitly mention supporting covariates and fall short of PatchTST in long-horizon benchmarks.

Recent work has improved the efficacy of RNNs (Kag et al., 2020; Lukoševičius and Uselis, 2022; Rusch and Mishra, 2020; Li et al., 2019b) and applied parameter efficient SSMs (Gu et al.; Gupta et al., 2022) to modeling long range dependencies in sequences. They have demonstrated improvement over some transformer based architectures on sequence modeling benchmarks including speech and 1-D pixel level image classification tasks. We compare our method to S4 model (Gu et al.), which is the only such method that has been applied to global univariate and global multivariate forecasting.

Note that all categories of models can be fairly compared on the task of multivariate long-term forecasting if they are evaluated on the same test set for the same task, which is the protocol that we follow in Section 5.

## 3    Problem Setting

Before we describe the problem setting we will need to setup some general notation.

### 3.1    Notation

We will denote matrices by bold capital letters like $\boldsymbol{X} \in \mathbb{R}^{N \times T}$. The slice notation $i : j$ denotes the set $\{i, i+1, \cdots j\}$ and $[n] := \{1, 2, \cdots, n\}$. The individual rows and columns are always treated as column vectors unless otherwise specified. We can also use sets to select sub-matrices i.e $\boldsymbol{X}[\mathcal{I}, \mathcal{J}]$ denotes the sub-matrix with rows in $\mathcal{I}$ and columns in $\mathcal{J}$. $\boldsymbol{X}[:, j]$ means selecting the $j$-th column while $\boldsymbol{X}[i, :]$ means the $i$-th row. The notation $[\boldsymbol{v}; \boldsymbol{u}]$ will denote the concatenation of the two column vectors and the same notation can be used for matrices along a dimension.

### 3.2    Multivariate Forecasting

In this section we first abstract out the core problem in long-term multivariate forecasting. There are $N$ time-series in the dataset. The look-back of the $i$-th time-series will be denoted by $\mathbf{y}_{1:L}^{(i)}$, while the horizon is denoted by $\mathbf{y}_{L+1:L+H}^{(i)}$. The task of the forecaster is to predict the horizon time-points given access to the look-back.

In many forecasting scenarios, there might be dynamic and static covariates that are known in advance. With slight abuse of notation, we will use $\boldsymbol{x}_t^{(i)} \in \mathbb{R}^r$ to denote the $r$-dimensional dynamic covariates of time-series $i$ at time $t$. For instance, they can be global covariates (common to all time-series) such as day of the week, holidays etc or specific to a time-series for instance the discount of a particular product on a particular day in a demand forecasting use case. We can also have static attributes of a time-series denoted by $\boldsymbol{a}^{(i)}$ such as

features of a product in retail that do not change with time. In many applications, these covariates are vital for accurate forecasting and a good model architecture should have provisions to handle them.

The forecaster can be thought of as a function that maps the history $\mathbf{y}_{1:L}^{(i)}$, the dynamic covariates $\boldsymbol{x}_{1:L+H}^{(i)}$ and the static attributes $\boldsymbol{a}^{(i)}$ to an accurate prediction of the future, i.e.,

$$f : \left( \left\{ \mathbf{y}_{1:L}^{(i)} \right\}_{i=1}^{N}, \left\{ \boldsymbol{x}_{1:L+H}^{(i)} \right\}_{i=1}^{N}, \left\{ \boldsymbol{a}^{(i)} \right\}_{i=1}^{N} \right) \longrightarrow \left\{ \hat{\mathbf{y}}_{L+1:L+H}^{(i)} \right\}_{i=1}^{N}. \tag{1}$$

The accuracy of the prediction will be measured by a metric that quantifies their closeness to the actual values. For instance, if the metric is Mean Squared Error (MSE), then the goodness of fit is measured by,

$$\text{MSE}\left( \left\{ \mathbf{y}_{L+1:L+H}^{(i)} \right\}_{i=1}^{N}, \left\{ \hat{\mathbf{y}}_{L+1:L+H}^{(i)} \right\}_{i=1}^{N} \right) = \frac{1}{NH} \sum_{i=1}^{N} \left\| \mathbf{y}_{L+1:L+H}^{(i)} - \hat{\mathbf{y}}_{L+1:L+H}^{(i)} \right\|_{2}^{2}. \tag{2}$$

## 4 Model

Recently, it has been observed that simple linear models (Zeng et al., 2023) can outperform Transformers based models in several long-term forecasting benchmarks. On the other hand, linear models will fall short when there are inherent non-linearities in the dependence of the future on the past. Furthermore, linear models would not be able to model the dependence of the prediction on the covariates as evidenced by the fact that (Zeng et al., 2023) do not use time-covariates as they hurt performance.

In this section, we introduce a simple and efficient MLP based architecture for long-term time-series forecasting. In our model we add non-linearities in the form of MLPs in a manner that can handle past data and covariates. The model is dubbed TiDE (Time-series Dense Encoder) as it encodes the past of a time-series along with covariates using dense MLP's and then decodes the encoded time-series along with future covariates.

An overview of our architecture has been presented in Figure 1. Our model is applied in a channel independent manner (the term was used in (Nie et al., 2022)) i.e the input to the model is the past and covariates of one time-series at a time $\left( \mathbf{y}_{1:L}^{(i)}, \boldsymbol{x}_{1:L}^{(i)}, \boldsymbol{a}^{(i)} \right)$ and it maps it to the prediction of that time-series $\hat{\mathbf{y}}_{L+1:L+H}^{(i)}$. Note that the weights of the model are trained globally using the whole dataset. A key component in our model is the MLP residual block towards the right of the figure.

**Residual Block.** We use the residual block as the basic layer in our architecture. It is an MLP with one hidden layer with ReLU activation. It also has a skip connection that is fully linear. We use dropout on the linear layer that maps the hidden layer to the output and also use layer norm at the output.

We separate the model into encoding and decoding sections. The encoding section has a novel feature projection step followed by a dense MLP encoder. The decoder section consists of a dense decoder followed by a novel temporal decoder. Note that the dense encoder (green block with $n_e$ layers) and decoder blocks (yellow block with $n_d$ layers) in Figure 1 can be merged into a single block. For the sake of exposition we keep them separate as we tune the hidden layer size in the two blocks separately. Also the last layer of the decoder block is unique in the sense that its output dimension needs to be $H \times p$ before the reshape operation.

### 4.1 Encoding

The task of the encoding step is to map the past and the covariates of a time-series to a dense representation of the features. The encoding in our model has two key steps.

**Feature Projection.** We use a residual block to map $\boldsymbol{x}_t^{(i)}$ at each time-step (both in the look-back and the horizon) into a lower dimensional projection of size $\tilde{r} \ll r$ (`temporalWidth`). This operation can be described

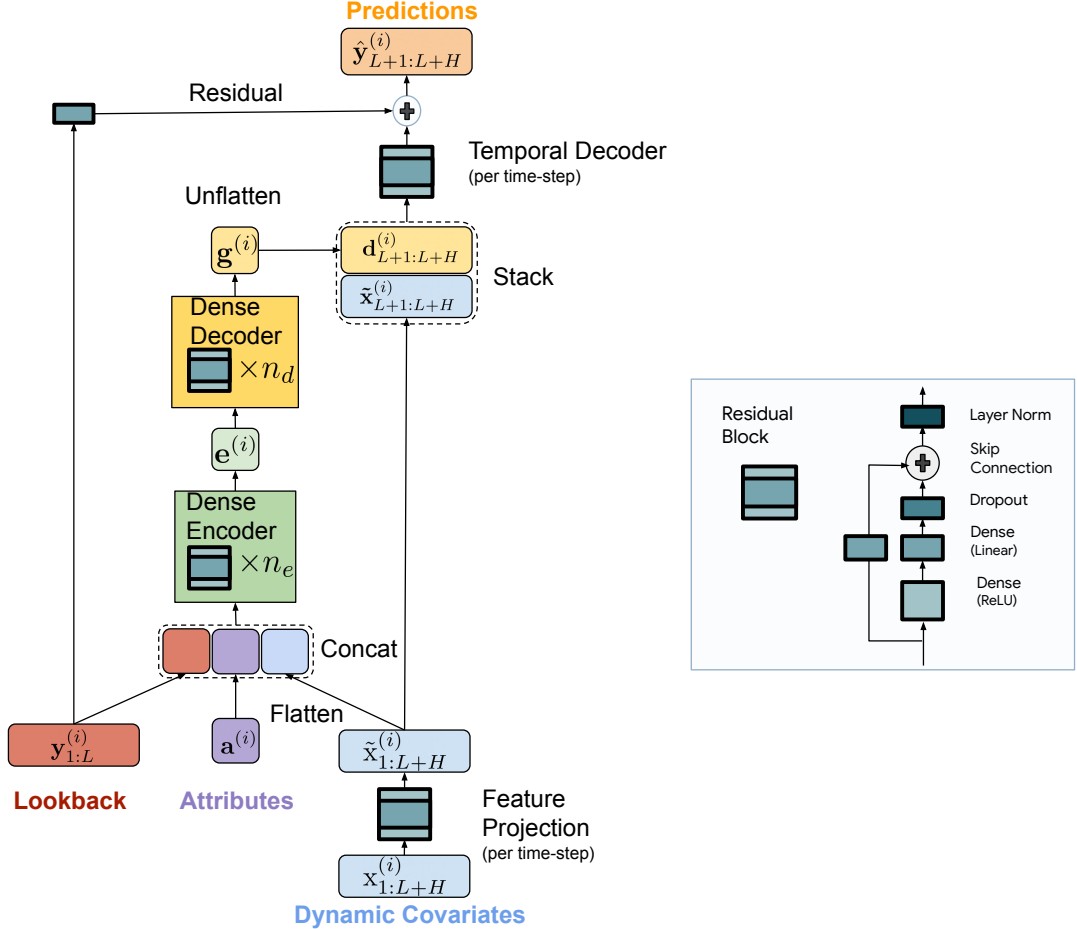

Figure 1: Overview of TiDE architecture. The dynamic covariates per time-point are mapped to a lower dimensional space using a feature projection step. Then the encoder combines the look-back along with the projected covariates with the static attributes to form an encoding. The decoder maps this encoding to a vector per time-step in the horizon. Then a temporal decoder combines this vector (per time-step) with the projected features of that time-step in the horizon to form the final predictions. We also add a global linear residual connection from the look-back to the horizon.

as,

$$\tilde{\boldsymbol{x}}_t^{(i)} = \text{ResidualBlock}\left(\boldsymbol{x}_t^{(i)}\right). \tag{3}$$

This is essentially a dimensionality reduction step since flattening the dynamic covariates for the whole look-back and horizon would lead to an input vector of size $(L + H)r$ which can be prohibitively large. On the other hand, flattening the reduced features would only lead to a dimension of $(L + H)\tilde{r}$.

**Dense Encoder.** As the input to the dense encoder, we stack and flatten all the past and future projected covariates, concatenate them with the static attributes and the past of the time-series. Then we map them to an embedding using an encoder which contains multiple residual blocks. This can be written as,

$$\boldsymbol{e}^{(i)} = \text{Encoder}\left(\mathbf{y}_{1:L}^{(i)}; \tilde{\boldsymbol{x}}_{1:L+H}^{(i)}; \boldsymbol{a}^{(i)}\right) \tag{4}$$

The encoder internal layer sizes are all set to `hiddenSize` and the total number of layers in the encoder is set to $n_e$ (`numEncoderLayers`).

## 4.2 Decoding

The decoding in our model maps the encoded hidden representations into future predictions of time series. It also comprises of two operations, dense decoder and temporal decoder.

**Dense Decoder.** The first decoding unit is a stacking of several residual blocks like the encoder with the same hidden layer sizes. It takes as an input the encoding $e^{(i)}$ and maps it to a vector $g^{(i)}$ of size $H \times p$ where $p$ is the `decoderOutputDim`. This vector is then reshaped to a matrix $D^{(i)} \in \mathbb{R}^{d \times H}$. The $t$-th column i.e $d_t^{(i)}$ can be thought of as the decoded vector for the $t$-th time-period in the horizon for all $t \in [H]$. This whole operation can be described as,

$$g^{(i)} = \text{Decoder}\left(e^{(i)}\right) \quad \in \mathbb{R}^{p.H}$$
$$D^{(i)} = \text{Reshape}\left(g^{(i)}\right) \quad \in \mathbb{R}^{p \times H}.$$

The number of layers in the dense decoder is $n_d$ (`numDecoderLayers`).

**Temporal Decoder.** Finally, we use the temporal decoder to generate the final predictions. The temporal decoder is just a residual block with output size 1 that maps the decoded vector $d_t^{(i)}$ at $t$-th horizon time-step concatenated with the projected covariates $\tilde{x}_{L+t}^{(i)}$ i.e

$$\hat{\mathbf{y}}_{L+t}^{(i)} = \text{TemporalDecoder}\left(d_t^{(i)}; \tilde{x}_{L+t}^{(i)}\right) \quad \forall t \in [H].$$

This operation adds a "highway" from the future covariates at time-step $L + t$ to the prediction at time-step $L + t$. This can be useful if some covariates have a strong direct effect on a particular time-step's actual value. For instance, in retail demand forecasting a holiday like Mother's day might strongly affect the sales of certain gift items. Such signals can be lost or take longer for the model to learn in absence of such a highway. We denote the hyperparameter controlling the hidden size of the temporal decoder as `temporalDecoderHidden`.

Finally, we add a *global residual connection* that linearly maps the look-back $\mathbf{y}_{1:L}^{(i)}$ to a vector the size of the horizon which is added to the prediction $\hat{\mathbf{y}}_{L+1:L+H}^{(i)}$. This ensures that a purely linear model like the one in (Zeng et al., 2023) is always a subclass of our model.

**Training and Evaluation.** The model is trained using mini-batch gradient descent where each batch consists of a `batchSize` number of time-series and the corresponding look-back and horizon time-points. We use MSE as the training loss. Each epoch consists of all look-back and horizon pairs that can be constructed from the training period i.e. two mini-batches can have overlapping time-points. This was standard practice in all prior work on long-term forecasting (Zeng et al., 2023; Liu et al., 2021; Wu et al., 2021; Li et al., 2019a).

The model is evaluated on a test set on every (look-back, horizon) pair that can be constructed from the test set. This is usually known as rolling validation/evaluation. A similar evaluation on a validation set can be used optionally used to tune parameters for model selection.

## 5 Experimental Results

In this section we present our main experimental results on popular long-term forecasting benchmarks. We also perform an ablation study that shows the usefulness of the temporal decoder.

### 5.1 Long-Term Time-Series Forecasting

**Datasets.** We use seven commonly used long-term forecasting benchmark datasets: Weather, Traffic, Electricity and 4 ETT datasets (ETTh1, ETTh2, ETTm1, ETTm2). We refer the reader to (Wu et al., 2021) for a detailed discussion on the datasets. In Table 1 we provide some statistics about the datasets. Note that Traffic and Electricity are the largest datasets with $> 800$ and $> 300$ time-series each having tens of thousands of time-points. Since we are only interested in long-term forecasting results in this section, we omit the shorter horizon ILI dataset.

| Dataset | #Time-Series | #Time-Points | Frequency |
|---------|--------------|--------------|-----------|
| Electricity | 321 | 26304 | 1 Hour |
| Traffic | 862 | 17544 | 1 Hour |
| Weather | 21 | 52696 | 10 Minutes |
| ETTh1 | 7 | 17420 | 1 Hour |
| ETTh2 | 7 | 17420 | 1 Hour |
| ETTm1 | 7 | 69680 | 15 Minutes |
| ETTm2 | 7 | 69680 | 15 Minutes |

Table 1: Summary of datasets.

**Baselines and Setup.** We choose SOTA Transformers based models for time-series including Fedformer (Zhou et al., 2022), Autoformer (Wu et al., 2021), Informer (Zhou et al., 2021), Pyraformer (Liu et al., 2021) and LongTrans (Li et al., 2019a). Recently, DLinear (Zeng et al., 2023) showed that simple linear models can outperform the above methods and therefore DLinear serves as an important baseline. We include N-HiTS (Challu et al., 2023) which is an improvement over the famous NBeats (Oreshkin et al.) model. Finally we compare with PatchTST (Nie et al., 2022) where they showed that vanilla Transformers applied to time-series patches can be very effective. The results for all Transformer based baselines are reported from (Nie et al., 2022).

For each method, the look-back window was tuned in $\{24, 48, 96, 192, 336, 720\}$. We report the DLinear numbers directly from the original paper (Zeng et al., 2023). For our method we always use context length of 720 for all horizon lengths in $\{96, 192, 336, 720\}$. All models were trained using MSE as the training loss. In all the datasets, the train:validation:test ratio is 7:1:2 as dictated by prior work. Note that all the experiments are performed on standard normalized datasets (using the mean and the standard deviations in the training period) in order to be consitent with prior work (Wu et al., 2021).

**Our Model.** We use the architecture described in Figure 1. We tune our hyper-parameters using the validation set rolling validation error. We provide details about our hyper-parameters in Appendix B.3. As global dynamic covariates, we use simple time-derived features like minute of the hour, hour of the day, day of the week etc which are normalized similar to (Alexandrov et al., 2020). Note that these features were turned off in DLinear since it was observed to hurt the performance of the linear model, however our model can easily handle such features. Our model is trained in Tensorflow (Abadi, 2016) and we optimize using the default settings of the Adam optimizer (Kingma and Ba, 2014). We provide our implementation in the supplementary with scripts to reproduce the results in Table 2.

**Results.** We present Mean Squared Error (MSE) and Mean Absolute Error (MSE) for all datasets and methods in Table 2. For our model we report the mean metric out of 5 independent runs for each setting. The bold-faced numbers are from the best model or within statistical significance of the best model in terms of two standard error intervals. Note that different predictive statistics are optimal for different target metrics (Awasthi et al., 2021; Gneiting, 2011) and therefore we should look at a target metric that is closely

---

[1]Note that there was a bug in the original test dataloader that affected the result significantly in smaller datasets like ETTh1 and ETTh2. We report the PatchTST results after correcting this bug in the dataloader.

| Models | | TiDE | | PatchTST/64 | | N-HiTS | | DLinear | | FEDformer | | Autoformer | | Informer | | Pyraformer | | LogTrans | |
|---|---|---|---|---|---|---|---|---|---|---|---|---|---|---|---|---|---|---|---|
| Metric | | MSE | MAE | MSE | MAE | MSE | MAE | MSE | MAE | MSE | MAE | MSE | MAE | MSE | MAE | MSE | MAE | MSE | MAE |
| Weather | 96 | 0.166 | 0.222 | **0.149** | **0.198** | 0.158 | **0.195** | 0.176 | 0.237 | 0.238 | 0.314 | 0.249 | 0.329 | 0.354 | 0.405 | 0.896 | 0.556 | 0.458 | 0.490 |
| | 192 | 0.209 | 0.263 | **0.194** | **0.241** | 0.211 | 0.247 | 0.220 | 0.282 | 0.275 | 0.329 | 0.325 | 0.370 | 0.419 | 0.434 | 0.622 | 0.624 | 0.658 | 0.589 |
| | 336 | 0.254 | 0.301 | **0.245** | **0.282** | 0.274 | 0.300 | 0.265 | 0.319 | 0.339 | 0.377 | 0.351 | 0.391 | 0.583 | 0.543 | 0.739 | 0.753 | 0.797 | 0.652 |
| | 720 | **0.313** | 0.340 | **0.314** | **0.334** | 0.401 | 0.413 | 0.323 | 0.362 | 0.389 | 0.409 | 0.415 | 0.426 | 0.916 | 0.705 | 1.004 | 0.934 | 0.869 | 0.675 |
| Traffic | 96 | **0.336** | 0.253 | 0.360 | **0.249** | 0.402 | 0.282 | 0.410 | 0.282 | 0.576 | 0.359 | 0.597 | 0.371 | 0.733 | 0.410 | 2.085 | 0.468 | 0.684 | 0.384 |
| | 192 | **0.346** | **0.257** | 0.379 | 0.256 | 0.420 | 0.297 | 0.423 | 0.287 | 0.610 | 0.380 | 0.607 | 0.382 | 0.777 | 0.435 | 0.867 | 0.467 | 0.685 | 0.390 |
| | 336 | **0.355** | **0.260** | 0.392 | 0.264 | 0.448 | 0.313 | 0.436 | 0.296 | 0.608 | 0.375 | 0.623 | 0.387 | 0.776 | 0.434 | 0.869 | 0.469 | 0.734 | 0.408 |
| | 720 | **0.386** | **0.273** | 0.432 | 0.286 | 0.539 | 0.353 | 0.466 | 0.315 | 0.621 | 0.375 | 0.639 | 0.395 | 0.827 | 0.466 | 0.881 | 0.473 | 0.717 | 0.396 |
| Electricity | 96 | **0.132** | 0.229 | **0.129** | **0.222** | 0.147 | 0.249 | 0.140 | 0.237 | 0.186 | 0.302 | 0.196 | 0.313 | 0.304 | 0.393 | 0.386 | 0.449 | 0.258 | 0.357 |
| | 192 | **0.147** | 0.243 | **0.147** | **0.240** | 0.167 | 0.269 | 0.153 | 0.249 | 0.197 | 0.311 | 0.211 | 0.324 | 0.327 | 0.417 | 0.386 | 0.443 | 0.266 | 0.368 |
| | 336 | **0.161** | 0.261 | 0.163 | 0.259 | 0.186 | 0.290 | 0.169 | 0.267 | 0.213 | 0.328 | 0.214 | 0.327 | 0.333 | 0.422 | 0.378 | 0.443 | 0.280 | 0.380 |
| | 720 | **0.196** | 0.294 | **0.197** | **0.290** | 0.243 | 0.340 | 0.203 | 0.301 | 0.233 | 0.344 | 0.236 | 0.342 | 0.351 | 0.427 | 0.376 | 0.445 | 0.283 | 0.376 |
| ETTh1 | 96 | **0.375** | **0.398** | 0.379 | 0.401 | 0.378 | **0.393** | **0.375** | **0.399** | 0.376 | 0.415 | 0.435 | 0.446 | 0.941 | 0.769 | 0.664 | 0.612 | 0.878 | 0.740 |
| | 192 | **0.412** | **0.422** | 0.413 | 0.429 | 0.427 | 0.436 | **0.412** | **0.420** | 0.423 | 0.446 | 0.456 | 0.457 | 1.007 | 0.786 | 0.790 | 0.681 | 1.037 | 0.824 |
| | 336 | **0.435** | **0.433** | **0.435** | 0.436 | 0.458 | 0.484 | 0.439 | 0.443 | 0.444 | 0.462 | 0.486 | 0.487 | 1.038 | 0.784 | 0.891 | 0.738 | 1.238 | 0.932 |
| | 720 | 0.454 | **0.465** | **0.446** | **0.464** | 0.472 | 0.561 | 0.501 | 0.490 | 0.469 | 0.492 | 0.515 | 0.517 | 1.144 | 0.857 | 0.963 | 0.782 | 1.135 | 0.852 |
| ETTh2 | 96 | **0.270** | **0.336** | 0.274 | 0.337 | 0.274 | 0.345 | 0.289 | 0.353 | 0.332 | 0.374 | 0.332 | 0.368 | 1.549 | 0.952 | 0.645 | 0.597 | 2.116 | 1.197 |
| | 192 | **0.332** | **0.380** | 0.338 | **0.376** | 0.353 | 0.401 | 0.383 | 0.418 | 0.407 | 0.446 | 0.426 | 0.434 | 3.792 | 1.542 | 0.788 | 0.683 | 4.315 | 1.635 |
| | 336 | **0.360** | 0.407 | 0.363 | **0.397** | 0.382 | 0.425 | 0.448 | 0.465 | 0.400 | 0.447 | 0.477 | 0.479 | 4.215 | 1.642 | 0.907 | 0.747 | 1.124 | 1.604 |
| | 720 | 0.419 | 0.451 | **0.393** | **0.430** | 0.625 | 0.557 | 0.605 | 0.551 | 0.412 | 0.469 | 0.453 | 0.490 | 3.656 | 1.619 | 0.963 | 0.783 | 3.188 | 1.540 |
| ETTm1 | 96 | 0.306 | 0.349 | **0.293** | **0.346** | 0.302 | 0.350 | 0.299 | 0.343 | 0.326 | 0.390 | 0.510 | 0.492 | 0.626 | 0.560 | 0.543 | 0.510 | 0.600 | 0.546 |
| | 192 | **0.335** | **0.366** | **0.333** | 0.370 | 0.347 | 0.383 | 0.335 | **0.365** | 0.365 | 0.415 | 0.514 | 0.495 | 0.725 | 0.619 | 0.557 | 0.537 | 0.837 | 0.700 |
| | 336 | **0.364** | **0.384** | 0.369 | 0.392 | 0.369 | 0.402 | 0.369 | 0.386 | 0.392 | 0.425 | 0.510 | 0.492 | 1.005 | 0.741 | 0.754 | 0.655 | 1.124 | 0.832 |
| | 720 | **0.413** | **0.413** | 0.416 | 0.420 | 0.431 | 0.441 | 0.425 | 0.421 | 0.446 | 0.458 | 0.527 | 0.493 | 1.133 | 0.845 | 0.908 | 0.724 | 1.153 | 0.820 |
| ETTm2 | 96 | **0.161** | **0.251** | 0.166 | 0.256 | 0.176 | 0.255 | 0.167 | 0.260 | 0.180 | 0.271 | 0.205 | 0.293 | 0.355 | 0.462 | 0.435 | 0.507 | 0.768 | 0.642 |
| | 192 | **0.215** | **0.289** | 0.223 | 0.296 | 0.245 | 0.305 | 0.224 | 0.303 | 0.252 | 0.318 | 0.278 | 0.336 | 0.595 | 0.586 | 0.730 | 0.673 | 0.989 | 0.757 |
| | 336 | **0.267** | **0.326** | 0.274 | 0.329 | 0.295 | 0.346 | 0.281 | 0.342 | 0.324 | 0.364 | 0.343 | 0.379 | 1.270 | 0.871 | 1.201 | 0.845 | 1.334 | 0.872 |
| | 720 | **0.352** | **0.383** | 0.362 | **0.385** | 0.401 | 0.413 | 0.397 | 0.421 | 0.410 | 0.420 | 0.414 | 0.419 | 3.001 | 1.267 | 3.625 | 1.451 | 3.048 | 1.328 |

Table 2: Multivariate long-term forecasting results with our model. $T \in \{96, 192, 336, 720\}$ for all datasets. The best results including the ones that cannot be statistically distinguished from the best mean numbers are in **bold**. We calculate standard error intervals for our method over 5 runs. The rest of the numbers are taken from the results from (Nie et al., 2022)[1]. All metrics are reported on standard normalized datasets. We provide the standard errors for our method in Table 5 in Appendix B.

aligned with the training loss in this case. Since all models were trained using MSE let us focus on that column for comparisons.

We can see that TiDE, PatchTST, N-HiTS and DLinear are much better than the other baselines in all datasets. This can be attributed to the fact that sub-quadratic approximations to the full self-attention mechanism is perhaps not best suited for long term forecasting. The same was observed in the PatchTST (Nie et al., 2022) where it was shown that full self attention over patches was much more effective even when applied in a channel dependent manner. For a more in depth discussion about the pitfalls of sub-quadratic attention approximation in the context of forecasting, we refer the reader to Section 3 of (Zeng et al., 2023). In Appendix A, we prove that a linear analogue of our model can be optimal for predicting linear dynamical systems when compared against sequence models, thus shedding some light on why our model and even simpler models like DLinear can be so competitive for long context and/or horizon forecasting.

Further, we outperform DLinear significantly in all settings except for horizon 192 in ETTh1 where the performances are equal. This shows the value of the additional non-linearity in our model. In some datasets like Weather and ETTh1, N-HiTS performs similar to TiDE and PatchTST for horizon 96, but fails to uphold the performance for longer horizons. In all datasets except Weather, we either outperform PatchTST or perform within its statistical significance for most horizons. In the Weather dataset, PatchTST peforms the best for horizons 96-336 while our model is the most performant for horizon 720. In the biggest dataset (Traffic), we significantly outperform PatchTST in all settings. For instance, for horizon 720 our prediction is 10.6% better than PatchTST in MSE. We provide additional results in Appendix B.1 including a comparison with the S4 model (Gu et al.) in Table 6.

## 5.2 Demand Forecasting

In order to showcase our model's ability to handle static attrubutes and complex dynamic covariates we use the M5 forecasting competition benchmarks (Makridakis et al., 2022). We follow the convention in the example notebook[2] released by the authors of (Alexandrov et al., 2020). The dataset consists of more than 30k time-series with static attributes like hierarchical categories and dynamic covariates like promotions. More details of the setup are available in Appendix B.1.

We present the competition metric (WRMSSE) results on the test set corresponding to the private leader-board in Table 3. We compare with DeepAR (Salinas et al., 2020) whose implementation can handle all covariates and also PatchTST (the best model from Table 2). Note that the implementation of PatchTST (Nie et al., 2022) does not handle covariates. We report the score over 3 independent runs along with the corresponding standard errors. We can see that PatchTST performs poorly as it does not use covariates. Our model using all the covariates outperforms DeepAR (that also uses all the covariates) by as much as 20%. For the sake of ablation, we also provide the metric for our model that uses only date derived features as covariates. There is a degradation in performance from not using the dataset specific covariates but even so this version of the model also outperforms the other baselines.

| Model | Covariates | Test WRMSSE |
|---|---|---|
| TiDE | Static + Dynamic | **0.611 ± 0.009** |
| TiDE | Date only | 0.637 ± 0.005 |
| DeepAR | Static + Dynamic | 0.789 ± 0.025 |
| PatchTST | None | 0.976 ± 0.014 |

Table 3: M5 forecasting results on the private test set. We report the competition metric (averaged across three runs) for each model. We also list the covariates used by all models.

## 5.3 Training and Inference Efficiency

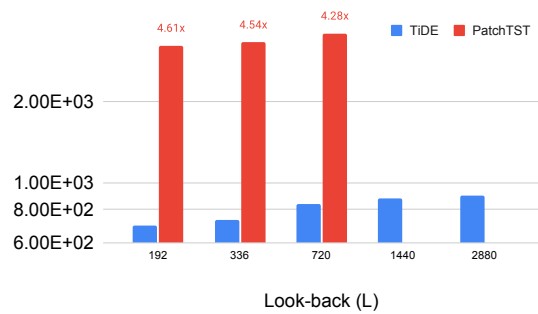

(a) Inference time per batch in microseconds

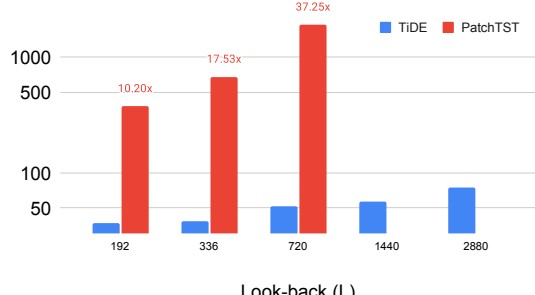

(b) Training time for one epoch in seconds.

Figure 2: In (a) we show the inference time per batch on the electricity dataset. In (b) we show the corresponding training times for one epoch. In both the figures the y-axis is plotted in log-scale. Note that the PatchTST model ran out of GPU memory for look-back $L \geq 1440$.

In the previous section we have seen that TiDE outperforms all methods except PatchTST by a large margin while it performs better or comparable to PatchTST in all datasets except Weather. Next, we would like to demonstrate that TiDE is much more efficient than PatchTST in terms of both training and inference times.

---

[2]`https://github.com/awslabs/gluonts/blob/dev/examples/m5_gluonts_template.ipynb`

Firstly, we would like to note that inference scales as $\tilde{O}(n_e h^2 + hL)$ for the encoder in TiDE, where $n_e$ is the number of layers in the encoder, $h$ is the hidden size of the internal layers and $L$ is the look-back. On the other hand, inference in PatchTST encoder would scale as $\tilde{O}(Kn_a L^2/P^2)$, where $K$ is the size of the key in self-attention, $P$ is the patch-size and $n_a$ is the number of attention layers. The quadratic scaling in $L$ can be prohibitive for very long contexts. Also, the amount of memory required is quadratic in $L$ for the vanilla Transformer architecture used in PatchTST [3].

We demonstrate these effects in practice in Figure 2. For the comparison to be fair we carry out the experiment using the data loader in the Autoformer (Wu et al., 2021) code base that was used in all subsequent papers. We use the electricity dataset with batch size 8, that is each batch has a shape of $8 \times 321 \times L$ because the electricity dataset has 321 time-series. We report the inference time for one batch and the training time for one epoch for TiDE and PatchTST as the look-back ($L$) is increased from 192 to 2880. We can see that there is an order of magnitude difference in inference time. The differences in training time is even more stark with PatchTST being much more sensitive to the look-back size. Further PatchTST runs out of memory for $L \geq 1440$. Thus our model achieves better or similar accuracy while being much more computation and memory efficient. All the experiments in this section were performed using a single NVIDIA T4 GPU on the same machine with 64 core Intel(R) Xeon(R) CPU @ 2.30GHz.

### 5.4 Ablation Study

**Temporal Decoder.** The use of the temporal decoder for adaptation to future covariates is perhaps one of the most interesting components of our model. Therefore in this section we would like to show the usefulness of that component with a semi-synthetic example using the electricity dataset.

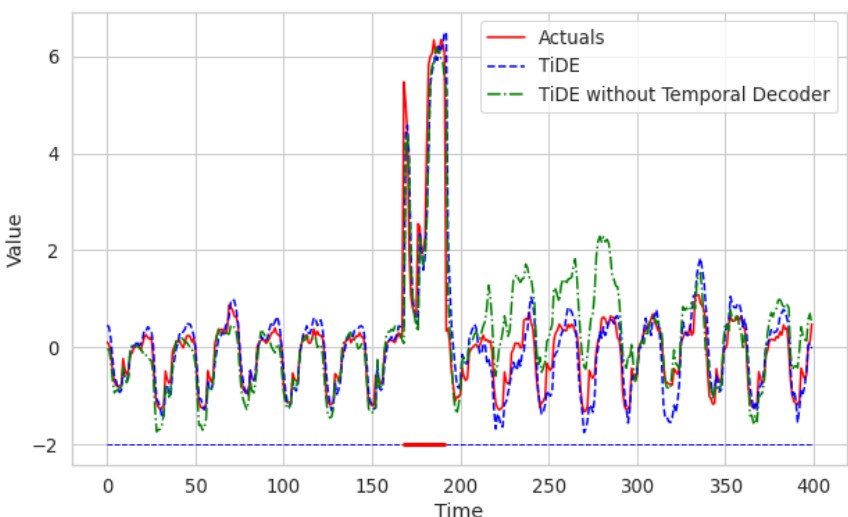

Figure 3: We plot the actuals vs the predictions from TiDE with and without the temporal decoder after just one epoch of training on the modified electricity dataset. The red part of the horizontal line indicates an event of Type A occuring.

We derive a new dataset from the electricity dataset, where we add numerical features for two kinds of events. When an event of Type A occurs the value of a time-series is increased by a factor which is uniformly chosen

---

[3]Note that sub-quadratic memory attention mechanism does exist (Dao et al., 2022) but has not been used in PatchTST. The quadratic computation seems to be unavoidable since approximations like Pyraformer seem to perform much worse than PatchTST.

between $[3, 3.2]$. When an event of Type B occurs the value of a time-series is decreased by a factor which is uniformly chosen between $[2, 2.2]$. Only 80% of the time-series are affected by these events and the time-series id's that fall in this bracket are chosen randomly. There are 4 numerical covariates that indicate Type A and 4 that indicate Type B. When Type A event occurs the Type A covariates are drawn from an isotropic Gaussian with mean $[1.0, 2.0, 2.0, 1.0]$ and variance 0.1 for every coordinate. On the other hand in the absence of Type A events Type A covariates are drawn from an isotropic Gaussian with mean $[0.0, 0.0, 0.0, 0.0]$. Thus these covariates serve as noisy indicators of the event. We follow a similar pattern for Type B events and covariates but with different means. Whenever these events occur they occur for 24 contiguous hours.

In order to showcase that the use of the temporal decoder can learn such patterns derived from the covariates faster, we plot the predictions from the TiDE model with and without the temporal decoder after just one epoch of training on the modified electricity dataset in Figure 3. The red part of the horizontal line indicates the occurrence of Type A events. We can see that the use of temporal decoder has a slight advantage during that time-period. But more importantly, in the time instances following the event the model without the temporal decoder is thrown off possibly because it has not yet readjusted its past to what it should have been without the event. This effect is negligible in the model which uses the temporal decoder, even after just one epoch of training.

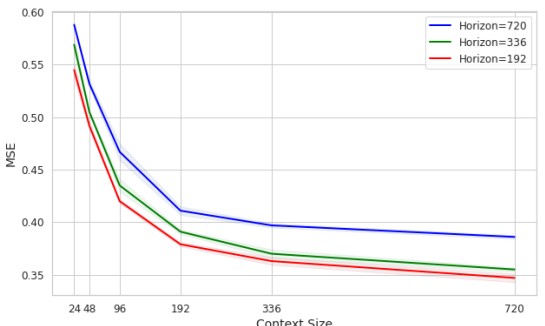

Figure 4: We plot the Test MSE on the traffic dataset as a function of different context sizes for three different horizon length tasks. Each plot is an average of 5 runs with the 2 standard error interval plotted.

| Models | | TiDE | | TiDE (no res.) | |
|---|---|---|---|---|---|
| Electricity | 96 | $\mathbf{0.132 \pm 0.003}$ | $\mathbf{0.229 \pm 0.001}$ | $0.136 \pm 0.001$ | $0.235 \pm 0.002$ |
| | 192 | $\mathbf{0.147 \pm 0.003}$ | $\mathbf{0.243 \pm 0.001}$ | $0.153 \pm 0.001$ | $0.253 \pm 0.001$ |
| | 336 | $\mathbf{0.161 \pm 0.001}$ | $\mathbf{0.261 \pm 0.002}$ | $0.172 \pm 0.003$ | $0.274 \pm 0.002$ |
| | 720 | $0.196 \pm 0.002$ | $0.294 \pm 0.001$ | $0.196 \pm 0.003$ | $0.295 \pm 0.002$ |

Figure 5: We perform an ablation study by presenting results from our model without any residual connections, on the electricity benchmark. We average over 5 runs for all the numbers and present the corresponding standard errors.

**Context Size.** In Figure 4 we study the dependence of prediction accuracy with context size on the Traffic dataset. We plot the results for multiple horizon length tasks and show that in all cases our methods performance becomes better with increasing context size as expected. This is contrast to some of the transformer based methods like Fedformer, Informer as shown in (Zeng et al., 2023).

**Residual Connections.** In Table 5 we perform an ablation study of the residual connections on the electricity dataset. In the model dubbed TiDE (no res) we remove all the residual connections including the ones in the residual block as well as the global linear residual connection. In horizons 96-336 we see a statistically significant drop in performance without the residual connections.

## 6 Conclusion

We propose a simple MLP based encoder decoder model that matches or supersedes the performance of prior neural network baselines on popular long-term forecasting benchmarks. At the same time, our model is 5-10x faster than the best Transformer based baselines. Our study shows that self attention might not be necessary to learn the periodicity and trend patterns at least for these long-term forecasting benchmarks.

Our theoretical analysis partly explains why this could be the case by proving that linear models can achieve near optimal rate when the ground truth is generated from a linear dynamical system. However, for future work it would be interesting to rigorously analyze MLPs and Transformer (including non-linearity) under some simple mathematical model for time-series data and potentially quantify the (dis)advantages of these architectures for different levels of seasonality and trends. Also, note that transformers are generally more parameter efficient than MLPs while being much more memory and compute intensive. This could be a limitation while training extremely large scale pre-trained models but the benefits of that line of work are not immediately clear in time-series forecasting and is beyond the scope of this work.

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

# A Theoretical Analysis under Linear Dynamical Systems

To gain insights into our design, we will now analyze the simplest linear analogue of our model. In our model if all the residual connections are active and the size of the encoding is greater than or equal to the length of the horizon, then it reduces to a linear map from the context and the covariates to the horizon. We study this version for the case when the data is generated from a Linear Dynamical System (LDS), that has been a popular mathematical model for systems evolving with time (Kalman, 1963). We will prove that under some conditions, a linear model that maps the past and the covariates of a finite context to the future can be optimal for prediction in a LDS.

## A.1 Theoretical Results

We formally define a linear dynamical system (LDS) as follows,

**Definition A.1.** A linear dynamical system (LDS) is a map from a sequence of input vectors $x_1, \ldots, x_T \in \mathbb{R}^n$ to output (response) vectors $y_1, \ldots, y_T \in \mathbb{R}^m$ of the form

$$h_{t+1} = Ah_t + Bx_t + \eta_t \tag{5}$$
$$y_t = Ch_t + Dx_t + \xi_t, \tag{6}$$

where $h_0, \ldots, h_T \in \mathbb{R}^d$ is a sequence of hidden states, $A, B, C, D$ are matrices of appropriate dimension, and $\eta_t \in \mathbb{R}^d, \xi_t \in \mathbb{R}^m$ are (possibly stochastic) noise vectors. The $x_t$'s can be thought of as covariates for the time-series $y_t$.

Given an LDS with parameter $\Theta = (A, B, C, D, h_0 = 0)$, we define the LDS predictor as follows,

**Definition A.2** (LDS predictor).

$$\hat{y}_t = y_{t-1} + (CB + D)x_t - Dx_{t-1} + \sum_{i=1}^{t-1} C(A^i - A^{i-1})Bx_{t-i} \tag{7}$$

For a fixed sequence length $T$, we consider a roll-out of the system $\{(x_t, y_t)\}_{t=1}^{T}$ to be a single example. In particular, we define $(X = (x_1, y_1, \ldots, x_{T-1}, y_{T-1}, x_T), Y = y_T)$ where $X$ contains be the full information that is available for the model to predict observation $y_T$. Trained on $N$ i.i.d. samples $\{(X_i, Y_i)\}$, the goal of the model is to predict $Y_i$ from $X_i$.

We assume the samples satisfy $\|x_t\|_2, \|y_t\|_2 \le c$ for a constant $c$. We compete against the class of functions in $\mathcal{H}$ restricted to contain LDSs with parameters $\Theta = (A, B, C, D, h_0 = 0)$ such that $0 \preccurlyeq A \preccurlyeq \gamma \cdot I$ where $\gamma < 1$ is a constant and $\|B\|_F, \|C\|_F, \|D\|_F \le c$ for an absolute constant $c$. For error metrics, we consider squared loss function $\ell_{X,Y} = \|h(x_1, y_1, \ldots, x_{T-1}, y_{T-1}, x_T) - y_T\|^2$. For an empirical sample set $S$, let $\ell_S(h) = \frac{1}{|S|} \sum_{(X,Y) \in S} \ell_{X,Y}(h)$. Similarly, for a distribution $\mathcal{D}$, let $\ell_{\mathcal{D}}(h) = \mathbb{E}_{(X,Y) \sim \mathcal{D}}[\ell_{X,Y}(h)]$.

Now we define our auto-regressive hypothesis class. Let $\tilde{X} \in \mathbb{R}^{(k+1)n+m}$ be the concatenated vector

$$\begin{bmatrix} x_{t-k} & x_{t-k+1} & \cdots & x_{t-1} & x_t & y_{t-1} \end{bmatrix},$$

and $f_M(\tilde{X}) = M\tilde{X}, M \in \mathbb{R}^{m \times (k+1)n+m}$. Our hypothesis class is defined as $\hat{\mathcal{H}} = \{f_M | \|M\|_F \le O(1)\}$.

We are ready to state our main theorem.

**Proposition A.3** (Generalization bound of learning LDS with auto-regressive algorithm). *Choose any $\varepsilon > 0$. Let $S = \{(X_i, Y_i)\}_{i=1}^{N}$ be a set of i.i.d. training samples from a distribution $\mathcal{D}$. Let $\hat{h} := \operatorname{argmin}_{h \in \hat{\mathcal{H}}} \ell_S(h)$ with a choice of $k = \Theta(\log(1/\varepsilon))$. Let $h^* := \operatorname{argmin}_{h^* \in \mathcal{H}} \ell_{\mathcal{D}}(h)$ be the loss minimizer over the set of LDS predictors. Then, with probability at least $1 - \delta$, it holds that*

$$\ell_{\mathcal{D}}(\hat{h}) - \min_{h \in \mathcal{H}} \ell_{\mathcal{D}}(h) \le \varepsilon + \frac{O\left(\log(1/\varepsilon)\sqrt{\log 1/\delta}\right)}{\sqrt{N}}.$$

The above result shows that the linear autoregressive predictor with a short look-back window is competitive against the best LDS predictor where the largest eigenvalue of the transition matrix $A$ is strictly smaller than 1. In Appendix A.2 we compare linear models with LSTM's and Transformers on long-term forecasting tasks on data generated by LDS, thus validating our theoretical results.

*Proof of Proposition A.3.* The proof proceeds as follows: First we show that an auto-regressive model with look by window length $\Omega(\log(1/\varepsilon))$ can approximate an LDS with error $\varepsilon$. Second, we prove a simple bound on the Rademacher complexity of the class of auto-regressive model we considered, which implies a generalization bound of our algorithm. Combining both results yields our main result.

**Proposition A.4** (Approximating LDS with auto-regressive model). *Let $\hat{y}_T$ be the predictions made by an LDS $\Theta = (A, B, C, D, h_0 = 0)$. Then, for any $\varepsilon > 0$, with a choice of $k = \Omega(\log(1/\varepsilon))$, there exists an $M_\Theta \in \mathbb{R}^{m \times (k+1)n+m}$ such that*

$$\left\| M_\Theta \tilde{X} - y_T \right\|^2 \leq \| \hat{y}_T - y_T \|^2 + \varepsilon.$$

*Proof.* This proposition is an analog of Theorem 3 in (Hazan et al., 2017). We construct $M_\Theta$ as the block matrix

$$\begin{bmatrix} M^{(k-1)} & M^{(k-2)} & \cdots & M^{(1)} & M^{(x')} & M^{(x)} & M^{(y)} \end{bmatrix},$$

where the blocks' dimensions are chosen to align with $\tilde{X}_t$, the concatenated vector

$$\begin{bmatrix} x_{t-k} & x_{t-k+1} & \cdots & x_{t-1} & x_t & y_{t-1} \end{bmatrix},$$

so that the prediction is the block matrix-vector product

$$M_\Theta \tilde{X}_t = \sum_{j=1}^{k-1} M^{(j)} x_{t-1-j} + M^{(x')} x_{t-1} + M^{(x)} x_t + M^{(y)} y_{t-1}.$$

Our construction is as follows:

- $M^{(j)} = C(A^{j+1} - A^j)B$, for each $1 \leq j \leq k-1$.

- $M^{(x')} = C(A - I)B - D, \quad M^{(x)} = CB + D, \quad M^{(y)} = I_{m \times m}.$

The prediction of the LDS is by definition

$$\hat{y}_t = y_{t-1} + (CB + D)x_t - Dx_{t-1} + \sum_{i=1}^{t-1} C(A^i - A^{i-1})Bx_{t-i}$$

We conclude that

$$\hat{y}_t = M_\Theta \tilde{X}_t + \sum_{i=k+2}^{T} C(A^i - A^{i-1})Bx_{t-i}.$$

This implies

$$\left\| M_\Theta \tilde{X}_t - y_t \right\|^2$$

$$\leq \| \hat{y}_t - y_t \|^2 + 2\| \hat{y}_t - y_t \| \left\| \sum_{i=k+2}^{T} C(A^i - A^{i-1})Bx_{t-i} \right\| + \left\| \sum_{i=k+2}^{T} C(A^i - A^{i-1})Bx_{t-i} \right\|^2$$

$$\leq \| \hat{y}_t - y_t \|^2 + O(\frac{\gamma^{k+2} + \gamma^{2k+4}}{1 - \gamma}) \leq \| \hat{y}_t - y_t \| + \varepsilon$$

$\square$

The empirical Rademacher complexity of $\hat{\mathcal{H}}$ on $N$ samples, with this restriction that $\|M\| = O(1)$, satisfies

$$\mathcal{R}_N(\hat{\mathcal{H}}) \leq O\left(\frac{1}{\sqrt{N}}\right).$$

It's easy to check that $\|M_\Theta\|_F \leq O\left(\sqrt{\sum_{i=0}^{k} \gamma^i}\right) = O(1)$, which falls in the feasible set of our algorithm. The maximum loss $\ell_{\max}$ of the hypothesis in model class $\hat{\mathcal{H}}$ is bounded by $O(k)$. The lipschitz constant of loss function in the matrix $M$ is $G_{\max} \leq \left\|M\tilde{X}_t - y_t\right\|_2 \cdot \left\|\tilde{X}_t\right\|_2 \leq O(k)$

With all of these facts in hand, a standard Rademacher complexity-dependent generalization bound holds in the improper hypothesis class $\hat{\mathcal{H}}$ (see, e.g. (Bartlett and Mendelson, 2002)):

**Lemma A.5** (Generalization via Rademacher complexity). *With probability at least $1 - \delta$, it holds that*

$$\ell_\mathcal{D}(\hat{h}) - \ell_\mathcal{D}(\hat{h}^*) \leq G_{\max}\mathcal{R}_N(\hat{\mathcal{H}}) + \ell_{\max}\sqrt{\frac{8\ln 2/\delta}{N}}$$

With the stated choice of $k$, an upper bound for the RHS of Lemma A.5 is

$$\frac{O\left(\log(1/\varepsilon)\sqrt{\log 1/\delta}\right)}{\sqrt{N}}.$$

Combining this with the approximation result (Proposition A.4) yields the theorem. □

## A.2 Experimental Results on Synthetic Datasets

**Dataset.** We evaluate several models on a synthetic dataset generated from a linear dynamical system. The transition matrix $A$ is a $30 \times 30$ dimension Wishart random matrix normalized to have operator norm equals 0.95. The noise $\eta_t$ in the state transition follows from a 30-dimensional Gaussian distribution. The input at each time-step $x_t$ follows from a 5-dimensional standard Gaussian distribution, which is observable to the model. Finally, We add seasonality of 6 different periodicity by adding cosine signal as the input to the linear dynamical system, but hidden from the prediction model. We generate 4 different time series which shares the same model parameter, input $x_t$ and seasonality input, with the only difference coming from the randomness in the state transition. We set look-back window to be length 320, and horizon also to length 320. For each time-series, we use the first 1640 steps for training, next 740 steps for validation, and the final 740 steps for testing, which results in 4000 examples for training, 400 examples for validation, and 400 examples for testing.

**Baselines and Setup.** We evaluate three models on our synthetic dataset: linear, long short-term memory (LSTM) and Transformer. Our linear model is a direct linear map between the history in look-back window and future. We use a one layer LSTM with dimension 128. For Transformer, we use a two layer self-attention layers with dimension 128, combined with a one hidden layer feed-forward network with 128 hidden units.

**Results.** We present Mean Squared Error (MSE) for all models in Table 4. For all models, we report the mean and standard deviation out of 3 independent runs for each setting. The bold-faced numbers are from the best model or within statistical significance of the best model in terms of two standard error intervals. We also plot the actuals (ground truth) vs the predictions from Linear, LSTM and Transformer models. We see that Transformer captures the lower frequency seasonality of the time-series but seems to not be able to leverage the inputs/covariates to predict the short term variation of the value, LSTM seems to not capture the trend/seasonality correctly, while Linear model's prediction is the closest to the truth, which matches the metric result in Table 4.

| Model | MSE |
|---|---|
| Linear | **0.510 ± 0.001** |
| LSTM | 1.455 ± 0.455 |
| Transformer | 0.731 ± 0.041 |

Table 4: Mean Squared Error (MSE) of Linear, LSTM and Transformer models on synthetic time-series.

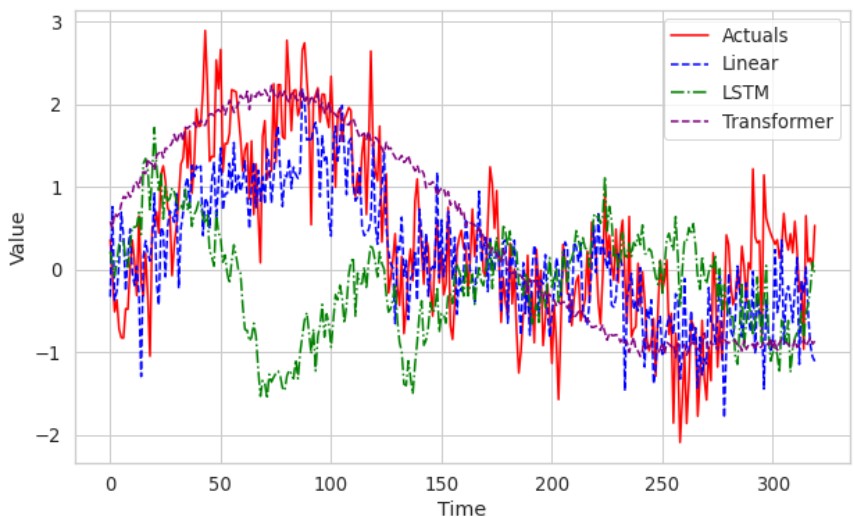

Figure 6: We plot the actuals vs the predictions from Linear, LSTM and Transformer models.

## B  More Experimental Details

### B.1  Additional Experiments

**Comparison against S4.**   In Table 6 we present the results of our model along side that of the S4 model (Gu et al.). The numbers of S4 are directly taken from Table-14 of the original paper. It can be seen that TiDE vastly outperforms the S4 model on the time-series benchmarks.

**M5 Forecasting.**   We will now provide more details about the M5 forecasting experiments. We follow the setup used in the notebook linked in Section 5 that was released by the authors of (Alexandrov et al., 2020). The list of dynamic features include date derived features, promotion features like snap_CA, snap_TX, snap_WI, even_type_1 and event_type_2. It also includes static attributes like category_id, store_id, department_id and item_id. The categorical features are embedded into learnable embeddings.

DeepAR (Salinas et al., 2020) has suggested using the zero-inflated negative binomial loss likelihood as the loss function for sparse count data in the dataset. Therefore we use this loss function for our model and the DeepAR model. All models are trained to a maximum of 100 epochs with an early stopping patience of 5.

### B.2  Data Loader

Each training batch consists of a look-back $\boldsymbol{Y}[\mathcal{B}, t - L : t - 1]$ and a horizon $\boldsymbol{Y}[\mathcal{B}, t : t + H - 1]$. Here, $t$ can range from $L + 1$ to $H$ steps before the end of the training set. $\mathcal{B}$ denotes the indices of the time-series in

| Models | | TiDE | |
|---|---|---|---|
| Electricity | 96 | $0.132 \pm 0.003$ | $0.229 \pm 0.001$ |
| | 192 | $0.147 \pm 0.003$ | $0.243 \pm 0.001$ |
| | 336 | $0.161 \pm 0.001$ | $0.261 \pm 0.002$ |
| | 720 | $0.196 \pm 0.002$ | $0.294 \pm 0.001$ |
| Traffic | 96 | $0.336 \pm 0.001$ | $0.253 \pm 0.001$ |
| | 192 | $0.346 \pm 0.001$ | $0.257 \pm 0.002$ |
| | 336 | $0.355 \pm 0.001$ | $0.260 \pm 0.001$ |
| | 720 | $0.386 \pm 0.002$ | $0.273 \pm 0.0005$ |
| Weather | 96 | $0.166 \pm 0.0005$ | $0.222 \pm 0.0005$ |
| | 192 | $0.209 \pm 0.002$ | $0.263 \pm 0.0001$ |
| | 336 | $0.254 \pm 0.002$ | $0.301 \pm 0.0001$ |
| | 720 | $0.313 \pm 0.001$ | $0.340 \pm 0.0002$ |
| ETTm2 | 96 | $0.161 \pm 0.0002$ | $0.251 \pm 0.0003$ |
| | 192 | $0.215 \pm 0.0001$ | $0.289 \pm 0.0004$ |
| | 336 | $0.267 \pm 0.0001$ | $0.326 \pm 0.0002$ |
| | 720 | $0.352 \pm 0.0002$ | $0.383 \pm 0.0002$ |
| ETTm1 | 96 | $0.306 \pm 0.0001$ | $0.349 \pm 0.0002$ |
| | 192 | $0.335 \pm 0.0002$ | $0.366 \pm 0.0002$ |
| | 336 | $0.364 \pm 0.0004$ | $0.384 \pm 0.0001$ |
| | 720 | $0.413 \pm 0.0001$ | $0.413 \pm 0.0001$ |
| ETTh1 | 96 | $0.375 \pm 0.0003$ | $0.398 \pm 0.0002$ |
| | 192 | $0.412 \pm 0.0002$ | $0.422 \pm 0.0001$ |
| | 336 | $0.435 \pm 0.0001$ | $0.433 \pm 0.0001$ |
| | 720 | $0.454 \pm 0.0003$ | $0.465 \pm 0.0001$ |
| ETTh2 | 96 | $0.270 \pm 0.0005$ | $0.336 \pm 0.0007$ |
| | 192 | $0.332 \pm 0.001$ | $0.380 \pm 0.002$ |
| | 336 | $0.360 \pm 0.001$ | $0.407 \pm 0.001$ |
| | 720 | $0.419 \pm 0.005$ | $0.451 \pm 0.002$ |

Table 5: We provide standard error bars for our method over 5 independent runs.

| Models | | TiDE | | S4 | |
|---|---|---|---|---|---|
| Metric | | MSE | MAE | MSE | MAE |
| Weather | 336 | 0.254 | 0.301 | 0.531 | 0.539 |
| | 720 | 0.313 | 0.340 | 0.578 | 0.578 |
| ETTh1 | 336 | 0.435 | 0.433 | 1.407 | 0.910 |
| | 720 | 0.454 | 0.465 | 1.162 | 0.842 |
| ETTh2 | 336 | 0.360 | 0.407 | 0.531 | 0.539 |
| | 720 | 0.419 | 0.451 | 2.650 | 1.340 |
| Electricity | 336 | 0.161 | 0.261 | 0.531 | 0.539 |
| | 720 | 0.196 | 0.294 | 0.578 | 0.578 |

Table 6: We benchmark our model's performance against that of S4. The S4 results are taken from Table 14 of the original paper (Gu et al.).

the batch and the `batchSize` can be set as a hyper-parameter. When `batchSize` is greater than $N$, all the time-series are loaded in a batch.

We also load time derived features as covariates. The time-stamps corresponding to time-indices $t - L : t + H - 1$ are converted to periodic features like minute of the hour, hour of the day, day of the week etc normalized to the scale $[-0.5, 0.5]$ as done in GluonTS (Alexandrov et al., 2020). In total we have 8 such features, many of which can stay constant depending on the granularity of the dataset.

### B.3 Hyperparameters

Recall that in Section 4, we had the following hyper-parameters `temporalWidth`, `hiddenSize`, `numEncoderLayers`, `numDecoderLayers`, `decoderOutputDim` and `temporalDecoderHidden`. We also have hyper-parameters `layerNorm` and `dropoutLevel` that denote the global model level layer norm on/off and the probability of dropout. We also tune the maximum `learningRate` which is the input to a cosine decay learning rate schedule. In all our experiments `batchSize` is fixed to 512 and `temporalWidth` is fixed to 4. We also tune whether reversible instance normalization (Kim et al., 2021) is turned on or off. The tuning range of the hparams are provided in Table 7. We use the validation loss to tune the hyper-parameters per dataset.

| Parameter | Range |
|---|---|
| hiddenSize | [256, 512, 1024] |
| numEncoderLayers | [1, 2, 3] |
| numDecoderLayers | [1, 2, 3] |
| decoderOutputDim | [4, 8, 16, 32] |
| temporalDecoderHidden | [32, 64, 128] |
| dropoutLevel | [0.0, 0.1, 0.2, 0.3, 0.5] |
| layerNorm | [True, False] |
| learningRate | Log-scale in [1e-5, 1e-2] |
| revIn | [True, False] |

Table 7: Ranges of different hyper-paramaters

We report the specific hyper-parameters chosen for each dataset in Table 8.

| Dataset | hiddenSize | numEncoderLayers | numDecoderLayers | decoderOutputDim | temporalDecoderHidden | dropoutLevel | layerNorm | learningRate | revIn |
|---|---|---|---|---|---|---|---|---|---|
| Traffic | 256 | 1 | 1 | 16 | 64 | 0.3 | False | 6.55e-5 | True |
| Electricity | 1024 | 2 | 2 | 8 | 64 | 0.5 | True | 9.99e-4 | False |
| ETTm1 | 1024 | 1 | 1 | 8 | 128 | 0.5 | True | 8.39e-5 | False |
| ETTm2 | 512 | 2 | 2 | 16 | 128 | 0.0 | True | 2.52e-4 | True |
| ETTh1 | 256 | 2 | 2 | 8 | 128 | 0.3 | True | 3.82e-5 | True |
| ETTh2 | 512 | 2 | 2 | 32 | 16 | 0.2 | True | 2.24e-4 | True |
| Weather | 512 | 1 | 1 | 8 | 16 | 0.0 | True | 3.01e-5 | False |

Table 8: The hyper-parameters for different experimental settings.

