# OpenReview forum: "Long-term Forecasting with TiDE: Time-series Dense Encoder"
_TMLR — Accepted by TMLR_

### Review · Reviewer_qgTo · 2023-06-08

**Summary Of Contributions:**

The paper presents an all-MLP model for time-series forecasting called TiDE. First, they apply a linear projection to reduce the input dimensionality of the time-series (independently for each time step). Then, they flatten all the inputs and they apply a stack of MLP residual blocks. Finally, they stack the initial (reduced) features to a reshaped output, and they apply a decoding step (also an MLP) to get the predictions. They compare TiDE to alternative transformer-based approachs, achieving competitive results.

**Audience:**

Yes

**Claims And Evidence:**

No

**Requested Changes:**

Concerning the comments above, I have the following requests which I think are critical right now:

1. Adding, if feasible, experiments where all potential inputs are used, including static attributes.
2. Clarify what is the practical difference between the encoder and the decoder.
3. Discuss more in-depth possible limitations of the method, and maybe related works concerning MLPs used in time-series forecasting (they are mostly discussing transformer-based models).
4. Either remove Section 5 or clarify what is the relation of TiDE with a linear model.

In addition, I have the following comments / observations which are of minor importance:

1. Fig. 1 needs some editing, especially concerning the proper use of padding (see left side of the yellow square), arrows (see Skip Connection), etc.
2. The paper has many sentences that clearly refer to the implementation, including "*the dropout and the layer norm can be turned on or off by a model level hyperparameter*", or the use of typewrite fonts for some variables (e.g., temporalWidth). I would suggest to remove these.
3. There are some typos, e.g., "sizer $H.p$". Also, why "." for multiplication?
4. $h_t$ in Eq. 6 should be $h_{t+1}$.
5. "*We constrain the comparator class*": what is the comparator class?
6. Inference time: what kind of implementation are you using for attention? Note that there are rather efficient implementations (e.g., FlashAttention) that can drastically improve results out-of-the-box.
7. Improvements compared to PatchTST (which is a basic transformer) are quite small. Also, 5 runs are too few to make statistical tests.
8. The authors are providing an analysis of the complexity, but if I understand correctly they are just considering the internal layers of the encoder. Shouldn't you also consider, e.g., the complexity of the feature projection step?

**Strengths And Weaknesses:**

The paper is relatively straightforward and easy to follow. Concerning the model, I fail to see much novelty in using MLPs for time-series forecasting, in particular:

1. They consider adding static attributes to the input (in violet in Fig. 1), but if I understand correctly this is not part of the experimental evaluation.

2. Maybe I also misunderstood this, but I don't see any reason of separating the "encoder" of the model from the "decoder" (green and yellow parts in Fig. 1), since they are both stacks of MLPs. They say the encoder has $n_e$ blocks and the decoder has $n_d$ blocks, but we could just as easily say that the first $n_e + 1$ blocks are the encoder and $n_d - 1$ are the decoder. Note that the authors are using "encoder-decoder" in the paper but only "encoder" in the name of the method.

Based on (1)-(2), the only major insight of the paper is to have an $Np$ output from the decoder, which is reshaped to $N \times p$ before applying another MLP row-wise for each prediction.

In addition, using MLPs has some drawbacks which are not highlighted here, for example, the model is forced to use a certain predefined lookback, and the number of parameters will grow linearly in the window size (both at the encoder input and at the decoder output).

I am also unsure about the meaning of Section 5. In section 5.1, they are proving that an autoregressive linear model is sufficient for modeling a linear dynamical system provided the state matrix satisfies some spectral constraints. Again, maybe I misunderstood something, but this seems a standard result in control theory (e.g., https://arxiv.org/pdf/1908.01039.pdf). I am also not sure how it compares to TiDE. They state that "*if all the residual connections are active and the size of the encoding is greater than or equal to the length of the horizon, then [our model] reduces to a linear map from the context and the covariates to the horizon.*". But how can this be true if there are ReLUs all over the place? By the same line, I am not really sure what they are trying to prove with the experiments in Section 5.2.

---

> ### Author Response · Authors · 2023-06-30
> **Thanks for the review.**
>
> We thank the reviewer for the detailed and thoughtful review. Below we address the questions and concerns grouped by topic:
>
> 1. __Encoder vs Decoder block:__ The reviewer is right that the encoder block in green with n_e layers can be conceptually merged with the decoder block with n_d layers. However, we keep these separate in the description because we can tune the hidden layer size in the encoder and the decoder blocks separately (two separate hyper-parameters). It is technically possible to tune all the (n_e + n_d) layer sizes as separate parameters but that would be too many hyper-parameters in practice. Also, as the reviewer has already noted, the last residual block of the decoder is special in the sense that it needs to have an output size of $p \times H$ that needs to be reshaped before feeding it into the temporal decoder. We have added a clarification of this to the paper.
>
> 2. __Major insight of the paper:__ From an architecture point of view we agree that the temporal decoder is novel. The handling of the covariate in both the encoder and decoder is also novel i.e the temporal projection per time-step followed by flattening and then passing it to the encoder.
>
> However, we would like to point out that one of the main points of the paper is that transformers are still not required to achieve SOTA performance on these popular long-term time-series benchmarks. Our all MLP architecture is both more efficient and performant, which we believe is a valuable insight for practitioners. We also explain the significance of our theoretical contributions below.
>
> 3. __Theoretical Contributions:__ As we mentioned in the beginning of section 5, “...if all the residual connections are active.., then it reduces to a linear map from the context and the covariates to the horizon.” We provide theoretical analysis of a simplified linear analogue of the TIDE model and prove that the linear model can achieve a near optimal error rate in a linear dynamical system. This also theoretically explains the effectiveness of approaches such as Dlinear (Zeng et al., 2023), which was not known before to the best of our knowledge.
>
> _The section on synthetic data experiment is not really very connected to the overall narrative of the architecture._
>
> Our synthetic experiment just validates the theory in section 5 that linear autoregressive models are competitive in the LDS settings.
>
> _Section 5 seems to be a standard result in control theory (e.g., https://arxiv.org/pdf/1908.01039.pdf)._
>
> It is indeed a standard result in control theory that an autoregressive–moving-average model with exogenous inputs (ARMAX) is model equivalent to a linear dynamical system. However, estimating ARMAX models is significantly harder than AR (autoregressive) models, and requires a different algorithm than a simple linear regression (https://arxiv.org/pdf/1908.01039.pdf). In fact, AR is strictly less expressive than LDS. What we showed in Section 5 is that a short look back AR model can approximate LDS with small approximation error, and is easy to learn.
>
> 4. __Experiments with covariates:__ We thank the reviewer for this suggestion. We have added a section on applying our model to the M5 forecasting competition which has dynamic covariates and static attributes. We also add DeepAR as a baseline that can handle both types of covariates. On the competition WRMSSE metric our model achieves a gain of 20% over DeepAR. We also add an ablation version of our model that does not use the dataset specific covariates. We hope these new results further demonstrate the effectiveness of our method.
>
> 5. __Related work and limitations:__ The reviewer is correct that transformers require less number of parameters than MLP models. We have added this in the conclusion section. However note that we are not in the realm of massively pretrained models where these considerations become more important. In our experiments, the MLP models actually have a lower memory footprint because the original transformer implementation needs a memory footprint quadratic in the context length. We have also added two prior MLP heavy models NBEATS and NHITS to the related work. Note that their performance on these benchmarks are worse than PatchTST.

---

> > ### Author Response · Authors · 2023-06-30
> > **Part 2 of rebuttal**
> >
> > __Minor Comments__
> >
> > 1. We have updated the figure to reflect this.
> >
> > 2. We have removed the quoted sentence.
> >
> > 3. Thanks for pointing them out, we have corrected them and some other typos.
> >
> > 4. Eq. 6 is actually correct as y_t depends on the current hidden state and not the future one.
> >
> > 5. We compete against the class of functions that are represented by the LDS parameters satisfying the conditions mentioned in that sentence. That is we measure our performance against the best function in that class in hindsight. We have changed the language in that sentence.
> >
> > 6. We have actually noted this point in footnote 2. The original implementation of PatchTST does not leverage flash attention. Note that flash attention obtains speedup using by modifying the computation to require fewer HBM access. However the orderwise computation complexity for context length remains quadratic so it will remain much slower than MLP models. Indeed the original flash attention paper claims a speed up of 1.1x - 3x in terms of training and inference speed. That would be insufficient to overcome the gap between our model and PatchTST in Figure 3.
> >
> > 7. The improvements are statistically significant wrt to the standard errors over 5 runs. Note that all prior baselines like PatchTST also report their metrics over 5 runs. We have added the standard error values for our model for all datasets and settings in the appendix. Also please note that improvement in accuracy is not the key differentiator, our model is as good or better than PatchTST while using significantly less compute and time.
> >
> > 8. The main point of our complexity analysis is to show that the dependence on the context length is linear, that is $O(L)$ while the vanilla transformers will require compute scaling as $O(L^2)$. The feature projection time complexity would just add a term of $O((rh + h\tilde{r})(L + H))$ which does not change the orderwise dependency on $L$.

---

### Review · Reviewer_w1d8 · 2023-06-17

**Summary Of Contributions:**

The authors propose a new MLP-based encoder-decoder model for long-term time-series forecasting that that combines the simplicity and speed of linear models while also being able to handle covariates and non-linear dependencies. They theoretically prove that their formulation obtains a near optimal rate for linear dynamical systems and empirical results demonstrate that their proposed algorithm matches or outperforms other benchmark algorithms on popular time-series benchmarks while having computational advantages in training and inference over state-of-the-art transformer models.

**Audience:**

Yes

**Claims And Evidence:**

Yes

**Requested Changes:**

Please address weaknesses mentioned above. In addition, could the authors analyze the weight matrices (or encoded representations) of their architecture to see if they are extracting any fundamental characteristics of the various data sets they examine? Have the authors also tried chaining multiple predictions from the output in a multistep prediction - do they observe any stability issues there?

**Strengths And Weaknesses:**

Strengths:
1. The proposed algorithm is simple, efficient, and the residual formulation connects to classical techniques in numerical methods for simulation dynamical systems.
2. The proposed algorithm is also competitive for various benchmarks and has computational benefits over more complex neural architectures.
3. The authors report their hyperparameter search parameters.

Weaknesses:
1. The ablation study is not thorough. The authors should widen the investigation to include the effect of the various components of the architecture, such the presence or absence of covariates, and the length of the lookback/horizon.
2. The metrics provided for comparison should also be provided with error bars to give a clearer picture of the performance of the algorithm.
3. The theoretical result related to the linear dynamical system seems a bit superfluous - can the authors connect this to a generic function approximation for time-series forecasting?
4. On a similar note - the section on synthetic data experiments does not really seem very connected to the overall narrative of the architecture.

---

> ### Author Response · Authors · 2023-06-30
> **Thanks for the review.**
>
> We thank the reviewer for the detailed review. Below we would like to address the main comments:
>
> 1.  We have now added two new ablation studies in Section 6.4. We show that our model’s performance becomes better monotonically with context size. We also show that our model performs better with residual connections than without it. Moreover we have added a whole new experiment on the M5 dataset with additional baselines. The m5 experiment has dataset specific static attributes and dynamical covariates that are important for forecasting accuracy. We show that our model can leverage these effects by adding an ablation version of our model that does not use these dataset specific covariates.
>
> 2. We actually compute error bars across 5 independent runs and only bold our numbers if they are statistically significantly the best (we mention this in Table 3). To address the reviewers comments, we have added a Table in the Appendix B.1 that explicitly reports our error bars.
>
> 3. As we mentioned in the beginning of section 5, “...if all the residual connections are active.., then it reduces to a linear map from the context and the covariates to the horizon.” We provide theoretical analysis of a simplified linear analogue of the TIDE model and prove that the linear model can achieve a near optimal error rate in a linear dynamical system. This also theoretically explains the effectiveness of approaches such as Dlinear (Zeng et al., 2023), which is not known before to the best of our knowledge.
> A full analysis of the non-linear parts of the architecture is beyond the scope of this paper and we have added a discussion about that in the conclusion section. However, please note that no prior work had a theoretical justification of why even the DLinear model might work better in long-term forecasting compared to sequence models.
>
>
> 4. The purpose of these experiments is to exactly validate our theoretical results. We show that linear models can beat LSTMS and Transformers when the ground truth is generated from an LDS.

---

### Review · Reviewer_FJaG · 2023-06-19

**Summary Of Contributions:**

This work proposes Time-Series Dense Encoder (TIDE) for solving the long-term time-series forecasting problem. TIDE is an MLP architecture.

It consists of the three inputs:
- (a) y_{1:L}   = past lookup (truth value on the past)
- (b) a         = attributes (common across time-series steps)
- (c) x_{1:L+H} = past observations (up to length L) and new observations (from L+1 to L+H)

It outputs the predictions for the ground truth y_{L+1:L+H}, i.e., the predictions for the time-horizon of length H starting at L+1.

TIDE architecture consists of many encoder layers followed by many decoder layers. These encoder/decoder layers follow the same residual block design consisting of skip connections and few feed-forward layers with non-linearities, dropout and layer-norm.

Empirical analysis shows that TIDE is 5-10x faster than earlier works on long-term time-series forecasting. This work also provides theoretical analysis of a simplied linear analogue of the TIDE model and proves that the linear model can achieve near optimal error rate in linear dynamical system.





**Audience:**

Yes

**Claims And Evidence:**

Yes

**Requested Changes:**


Questions for Authors:
------------

- Do LSTMs and other baselines utilize same attributes ( look-back truth values, attributes, etc. )

- Why do you not consider other RNN works that model long range dependencies?

- As the time-sequence progresses (L increases), the compute cost of the architectures increases since it sees both y_{1:L} and x_{1:L+H}. How much is the compute compared to LSTMs / Transformers?

- In Sec.4.Model: you say "Our model is applied in a channel independent manner.." what does it mean to say channel dependency or independency? Does it refer to some statistic that is common across many time-series?

- Figure.1: Is there any reasoning for the residual block? There could be many possibilities to design one for this use-case. Why did you choose this particular structure? In addition, why not other forms of non-linearities (instead of ReLU, LayerNorm, etc.) ?

- Sec.5.2: Why does LSTM only have one layer and transformer two layers in the synthetic experiments?

- Sec.6.2: Do these 5x and 10x efficiency numbers hold true for datasets other than Electricity?

- In many instances, you mentioned the use of Vanilla transformers (that would model N^2 dependencies on a sequence of length N), did you try out efficient linear or N log-N transformers?

- Do you have any ablations on various choices in the residual block? It is unclear why the proposed residual block design is the best suited for this task.

Missing Related Works:
---------

- Structured State Spaces
  - Efficiently Modeling Long Sequences with Structured State Spaces (https://arxiv.org/abs/2111.00396)
  - Diagonal State Spaces are as Effective as Structured State Spaces (https://arxiv.org/abs/2203.14343)

- ODE-RNNs / Orthogonal RNNs / Deep Linear RNNs / Deep RNNs
  - Independently Recurrent Neural Network (IndRNN): Building A Longer and Deeper RNN (https://arxiv.org/abs/1803.04831)
  - Deep Independently Recurrent Neural Network (IndRNN) (https://arxiv.org/abs/1910.06251)
  - CoRNN : https://arxiv.org/pdf/2010.00951.pdf
  - Time Adaptive RNNs (https://openaccess.thecvf.com/content/CVPR2021/papers/Kag_Time_Adaptive_Recurrent_Neural_Network_CVPR_2021_paper.pdf)
  - RNNs Incrementally Evolving on an Equilibrium Manifold: A Panacea for Vanishing and Exploding Gradients? (https://openreview.net/forum?id=HylpqA4FwS)

- Efficient Transformers that scale linearly in length of the input sequence (https://arxiv.org/abs/2009.06732), for instance: LinFormer and Long Short Transformer


Nit-Picks:
--------

- Sec.4.Model: ".. can outperform Tranformer..."
- Sec.6.Baselines and Setup: "We choose SOTA Tranformer..." (in particular check for the Transformer spelling in the paper, many places this is a spell error)
- Sec.7.Conclusion: "... our model if 5-10x ...."
- Figure.1: This should be aligned to the centered to improve the asthetics.
- In addition, please check for spell-errors and grammar in the paper. There seem to be many places where such issues exist.

**Strengths And Weaknesses:**



Strengths:
-----------
- TIDE seems to be able to model many interactions (look-back, direct feedback from the previous ground-truth, direct mapping from current observations into the output, interaction between attributes+look-back+dynamic-covariates) that are crucial from the long-term time-series forecasting perspective.

- Inference and training time improvements compared to the baselines.


Weaknesses:
-----------

- Theoretical analysis is for the linear dynamics and it is unclear how that relates to the proposed TIDE architecture (it has far too many non-linearities). This seems disconnected from the proposed TIDE architecture.

- Missing many related works for the long-term time-series prediction problems.

- While TIDE maps many necessary interactions required for long-term forecasting problems, it is unclear why a Transformer like architecture would fail on this task? Since, they can exploit N^2 paired dependencies betweeen the input covariats. In addition, if you simply concatenate lookup+attributes+observations, they ideally should be able to learn the desired dependencies.

- Missing evaluations of the other baselines that are well suited to the task at hand ( Efficient Transformers, Long-range dependencies ODE-RNNs, Structured State-space models )

---

> ### Author Response · Authors · 2023-06-30
> **Thanks for the review**
>
> We thank the reviewer for their detailed review and comments. Below we would like address the key points:
>
> 1.  __Theoretical analysis:__ As we mentioned in the beginning of section 5, “...if all the residual connections are active.., then it reduces to a linear map from the context and the covariates to the horizon.” We provide theoretical analysis of a simplified linear analogue of the TIDE model and prove that the linear model can achieve a near optimal error rate in a linear dynamical system. This also theoretically explains the effectiveness of approaches such as Dlinear (Zeng et al., 2023), which is not known before to the best of our knowledge.
> A full analysis of the non-linear parts of the architecture is beyond the scope of this paper and we have added a discussion about that in the conclusion section. However, please note that no prior work had a theoretical justification of why even the DLinear model might work better in long-term forecasting compared to sequence models.
>
> 2. __Long range RNN and related work:__ We thank the reviewer for pointing this out. We have added all the suggested works in the related work section. However, please note all the papers focus on sequence models that work well across multiple modalities (image, speech, text) and not specifically for time-series forecasting, most of them do not even benchmark on any forecasting datasets. The S4 (https://arxiv.org/abs/2111.00396) does benchmark on the same long-term forecasting dataset which is the focus of our work. Therefore we have added comparison to the S4 architecture in Table 5. Note that S4 performs worse than FedFormer and therefore is not competitive with our models or PatchTST.
>
> 3. __Transformers capturing dependencies:__ It is true that the vanilla transformers can capture $O(N^2)$ dependencies but note that all baselines such as FedFormer all the way to LongTrans use $O(N)$ or $O(N\log N)$ approximations for efficiency purposes which do not work well. Note that PatchTST which applies the vanilla attention on patches instead of single time-points does work relatively well (better than DLinear). The main point is that our model can achieve the same or better performance without any attention thereby being much more efficient in terms of compute and memory. This effectively shows that at least for these long terms forecasting benchmarks self attention is not needed to get SOTA performance.
>
> 4. __Efficient Transformers:__ The comparisons to efficient $O(N\log N)$ transformers is actually not missing. The works like LinFormer propose the approximations in a general setting. All the baselines (except PatchTST and DLinear) such as Fedformer, Informer, Autoformer, Pyraformer, LongTrans use these sub-quadratic approximations, specifically adapted to time-series forecasting applications. For instance, Pyraformer uses a pyramidal attention that is $O(N)$ where $N$ is the input length . So we do in fact compare to efficient transformers and in fact this shows that these sub-quadratic approximations do not work well in long-term forecasting and are easily beaten by simple linear models like DLinear. PatchTST showed that applying full self-attention but on patches is much more effective.

---

> > ### Author Response · Authors · 2023-06-30
> > **Part 2 of rebuttal**
> >
> > __Questions__
> >
> > 1.  All baselines are compared on the exact same task. Note that we have added new experiments on the M5 dataset which contains static and dynamic attributes that have a large  impact on the accuracy. We have also added comparisons to DeepAR that can also handle all these covariates.
> >
> > 2. We have discussed this above and have added a comparison to the S4 model.
> >
> > 3. The computation grows as $O(L)$ for our model but $O(L^2)$ for PatchTST as discussed both theoretically and empirically in Section 6.3 in the revised paper. LSTM can also maintain the $O(L)$ scaling but during training they need sequential roll out and therefore cannot take full advantage of GPU parallelism and thus end up actually being slower to train than even transformers.
> >
> > 4. Channel independence is a term that was used in the PatchTST paper to describe that the model maps each time-series's past and covariates to its future rather than mapping to the past of all the time-series to all the futures through a joint multivariate model.  We use the same terminology and  have also described it in the remaining part of the quoted sentence.
> >
> > 5. Residual connections is a standard trick now heavily used in deep learning to stabilize training by avoiding gradient explosion/collapse (for example in https://arxiv.org/pdf/1603.05027.pdf). We just add residual connection to a one hidden layer MLP to form the residual block. The hidden layer does use ReLU activation and we also have the option of enabling layer norm.
> >
> > 6. We did hyperparameter selection and one layer LSTM has the best performance, while two layer transformers have the best performance
> >
> > 7. Yes the efficiency numbers are independent of the dataset and just depend on the context and horizon lengths.
> >
> > 8. We have discussed this above and explained that many of the baselines have O(N \log N) approximations.
> >
> > 9. We thank the reviewer for the suggestion. We have added an ablation by removing all the residual connections in Section 6.4 of the revised paper. It shows that residual connections do help in improving the accuracy.
> >
> > Nit picks: We thank the reviewer for catching these. We have corrected them in the revision.

---

> > ### Comment · Reviewer_FJaG · 2023-07-28
> > **Response to Rebuttal**
> >
> > Thank you for addressing my comments. Please capture the essence of the O(N) or O(N log N) transformers used in the paper so that a reader can clearly understand why transformers (used in practice) fail on this task.

---

> > > ### Author Response · Authors · 2023-07-28
> > > **Thank you.**
> > >
> > > Thank you for your time in reviewing the paper and the great suggestion. We have added further discussion about the sub-quadratic approximations used in prior work both in the related work section and the long term forecasting results section in the latest revision.

---

### Review · Reviewer_5YuA · 2023-06-21

**Summary Of Contributions:**

The paper proposes a new time-series modeling tool and tests it on a series of short time-series modeling tasks.

**Audience:**

No

**Claims And Evidence:**

Yes

**Requested Changes:**

Based on the concerns I raised in the previous section, I vote for a major revision of the paper and encourage the authors to work on the suggestions made to make substantial improvements to their future work.

**Strengths And Weaknesses:**

The paper, unfortunately, misses heavily on including a series of state-of-the-art baseline models and datasets to test sequence modeling capabilities. More specifically, here is a non-exhaustive list of the SOTA models for capturing long-term dependencies:

[1] Gu, A., Goel, K., & Ré, C. (2021). Efficiently modeling long sequences with structured state spaces. arXiv preprint arXiv:2111.00396

[2] Gupta, A. (2022). Diagonal State Spaces are as Effective as Structured State Spaces. arXiv preprint arXiv:2203.14343.

[3] Gu, A., Johnson, I., Goel, K., Saab, K., Dao, T., Rudra, A., & Ré, C. (2021). Combining recurrent, convolutional, and continuous-time models with linear state space layers. Advances in neural information processing systems, 34, 572-585.

[4] Gu, A., Dao, T., Ermon, S., Rudra, A., & Ré, C. (2020). Hippo: Recurrent memory with optimal polynomial projections. Advances in Neural Information Processing Systems, 33, 1474-1487.

[5] Smith, J. T., Warrington, A., & Linderman, S. W. (2022). Simplified state space layers for sequence modeling. arXiv preprint arXiv:2208.04933.

[6] Hasani, R., Lechner, M., Wang, T.H., Chahine, M., Amini, A. and Rus, D., 2022. Liquid structural state-space models. arXiv preprint arXiv:2209.12951.

[7] Gu, A., Gupta, A., Goel, K. and Ré, C., 2022. On the parameterization and initialization of diagonal state space models. arXiv preprint arXiv:2206.11893.

[8] Gu, A., Johnson, I., Timalsina, A., Rudra, A. and Ré, C., 2022. How to train your hippo: State space models with generalized orthogonal basis projections. arXiv preprint arXiv:2206.12037

[9] Orvieto, Antonio, et al. "Resurrecting recurrent neural networks for long sequences." arXiv preprint arXiv:2303.06349 (2023).

[10] Ma, Xuezhe, et al. "Mega: moving average equipped gated attention." arXiv preprint arXiv:2209.10655 (2022).

[11] Morrill, James, et al. "Neural rough differential equations for long time series." International Conference on Machine Learning. PMLR, 2021.

[12] Rusch, T. Konstantin, et al. "Long expressive memory for sequence modeling." arXiv preprint arXiv:2110.04744 (2021).

The experiments on time-series forecasting are done on short-term sequences (below 720 steps) where even an ARIMA model, XGBoost, or a well-tuned LSTM could perform competitively.

There are established and challenging benchmarks with 8k-16k samples in the time-series domain. An example is the Speech Command recognition dataset used in all modern sequence modeling works. In this space, RNNs and State-space models work remarkably well. Another benchmark that must be tested is the long-range arena (LRA).

---

> ### Author Response · Authors · 2023-06-30
> **Rebuttal**
>
> We would like to note that all  the above papers focus on sequence models that work across multiple modalities (image, speech, text) and not specifically for time-series forecasting, and most of them (except for [1,3,6,7,8]) do not even benchmark on any forecasting datasets . In particular, among all the papers above only [1] performs benchmarks across the standard forecasting datasets typically used in the forecasting community (see the papers from Fedformer (Zhou et al., 2022), Autoformer (Wu et al., 2021), Informer (Zhou et al., 2021), Pyraformer (Liu et al., 2021) and LongTrans (Li et al., 2019), DLinear (Zeng et al., 2023),  PatchTST (Nie et al., 2022)). And the forecasting results reported in [1] (S4) are far from state-of-the-art - it performs even worse than reported numbers from DLinear, let alone the current-state-of-the-art (prior to our work) PatchTST model.  Also we have added a comparison to S4 using the numbers from Table. 14 of the original paper in Table. 5 of our revised paper.
>
> Possibly for the same reasons above, none of the papers mentioned by the reviewer have been cited in the recent forecasting papers (PatchTST, DLinear, Fedformer, etc) . We have added a discussion about this along with relevant citations in the related work section.
>
> We would also like to point out that benchmarking on Speech, LRA etc is outside the scope of our paper - we specifically focus on time-series forecasting.
>
> Regarding the reviewer’s statement “The experiments on time-series forecasting are done on short-term sequences (below 720 steps) where even an ARIMA model, XGBoost, or a well-tuned LSTM could perform competitively.”: this is incorrect (in fact, Table 13 from paper [1] listed by the reviewer (and also experiments in other papers such as Informer) already shows that that is not the case)

---

### Decision · Action_Editors · 2023-08-06

**Recommendation:** Accept with minor revision

**Comment:**

Thanks to the reviewers' comments, the current version of the paper is much more solid than the first submission. Notwithstanding two of the reviewers are not in favour of the acceptance of the submission, their arguments are either not applicable to time-series prediction tasks, or they concerns issues that can be fixed with a minor revision of the paper. The only issue that would hinder the acceptance of the submission would be the increase of the number of parameters per time-step. Given that nowadays we are witnessing models with trillions of parameters that can be used in practice, I personally believe this issue can be considered minor from a practical point of view. Of course, this does not mean that further research should not be performed to try to solve or reduce this issue. I believe, the contribution of the paper is interesting since it is showing that current Transformer-based models are, at the end of the day, not the best solution for time-series prediction tasks. This also holds for RNN-based solutions. The proposed model is inspired by work done using linear models, but by itself it is not a linear model. This is the main reason why all reviewers did not find the theoretical analysis (Section 5) so interesting. Authors justify the analysis as covering a very special case of the proposed architecture. I tend to agree with the reviewers, although I understand the author's argument. I suggest that the final version of the paper moves Section 5 in Appendix A, so to remove the confusion that it may generate in most of the readers. I also suggest to move at the very beginning of Section 4 the remark about the fact that the encoder and the decoder can be merged in a single block. Personally, I would also change the name to the Encoder and Decoder, since the functions that they implement are more related to the creation of an embedding, i.e. the real Encoder is the merge of the two blocks (current Encoder and Decoder), while the real Decoder is the Temporal Decoder.

**Audience:**

There will be individuals in TMLR's audience interested to the topic since time-series prediction is a very important task with applications in many areas.

**Claims And Evidence:**

The claims made in the submission are supported by accurate experiments and comparisons versus SOTA approaches for time-series prediction.

---

> ### Author Response · Authors · 2023-08-08
> **Thank you.**
>
> A big thank you to the area chair and the reviewers for taking the time to review the paper. The suggested changes have gone a long way to improve the paper.
>
> We have made the minor revision requested by the area chair:
>
> 1. We have moved the theory section to the appendix.
> 2. We have moved the clarification w.r.t encoder/decoder blocks towards the beginning of section 4.
> 3. We have also made minor changes to the architecture figure to change the name encode (decoder) -> dense encoder (dense decoder) as used in the revised text.
>
> Thanks again and the changes have been reflected in the camera ready version.